# Impact of the Over-the-Head Position with a Supraglottic Airway Device on Chest Compression Depth and Rate: A Post Hoc Analysis of a Randomized Controlled Trial

**DOI:** 10.3390/healthcare10040718

**Published:** 2022-04-13

**Authors:** Loric Stuby, Laurent Suppan, Laurent Jampen, David Thurre

**Affiliations:** 1Genève TEAM Ambulances, Emergency Medical Services, CH-1201 Geneva, Switzerland; d.thurre@gt-ambulances.ch; 2Division of Emergency Medicine, Department of Anaesthesiology, Clinical Pharmacology, Intensive Care and Emergency Medicine, Geneva University Hospitals, Faculty of Medicine, University of Geneva, CH-1211 Geneva, Switzerland; laurent.suppan@hcuge.ch; 3ESAMB—École Supérieure de Soins Ambulanciers, College of Higher Education in Ambulance Care, CH-1231 Conches, Switzerland; laurent.jampen@edu.ge.ch

**Keywords:** emergency medical services, paramedics, airway, supraglottic airway device, cardiac arrest, i-gel^®^, CPR, prehospital, resuscitation, chest compression depth

## Abstract

There is considerable controversy regarding the optimal airway management strategy in the case of out-of-hospital cardiac arrest. Registry-based studies yield contradicting results and the actual impact of using supraglottic devices on survival and neurological outcomes remains unknown. In a recent simulation study, the use of an i-gel^®^ device was associated with significantly shallower chest compressions. It was hypothesized that these shallower compressions could be linked to the provision of chest compressions in an over-the-head position, to the cumbersome airway management apparatus, and to a shallower i-gel^®^ insertion depth in the manikin. To test this hypothesis, we carried out a post hoc analysis, which is described in this report. Briefly, no association was found between the over-the-head position and compression depth.

## 1. Introduction

After cardiac arrest (CA), high-quality chest compressions, at a rate of 100 to 120 per minute with a depth of 5 to 6 cm, are required to increase the probability of achieving the return of spontaneous circulation (ROSC) [1]. We recently carried out a randomized controlled trial to determine whether the chest compression fraction (CCF) could be increased by using a supraglottic airway device (SGA) rather than a bag-valve-mask (BVM) device. We found that the i-gel^®^ allowed higher CCFs, and better ventilations parameters without delaying critical actions (time to first shock and time to first ventilation). However, we found that chest compressions were unexpectedly and significantly shallower in the SGA group (median (Q1; Q3) in the i-gel^®^ group 4.6 cm (4.3; 5.0) versus 5.2 cm (4.9; 5.3) in the BVM group, *p* = 0.007). Accordingly, the mean proportion of compressions within the recommended depth target (5 to 6 cm) was lower in the experimental group (41.7% (95% CI 28.2–55.3) versus 66.5% (95% CI 51.5–81.4), *p* = 0.01) [2,3]. Even though these results were obtained through a simulation study, their clinical impact is be worthy of attention. Indeed, several prior analyses of CA registries reported that the use of SGA devices was associated with lower ROSC rates [4,5,6]. These registry-based studies all present multiple biases, including a lack of data regarding chest compression depth and rate. Considering the shallower compressions reported in our simulation study, it is reasonable to believe that differences in chest compression depths could at least partly explain the lower rates of ROSC associated with the use of SGA devices in these registry-based studies.

Among the potential reasons that could explain the shallower compressions found in the SGA group, we proposed the hypothesis that the cumbersome airway management apparatus coupled with the shallower i-gel^®^ insertion depth in manikins (as compared to humans) may have prevented the adequate provision of chest compressions when paramedics used an over-the-head position [7]. Therefore, our objective is to determine whether chest compression depth is more adequate before i-gel^®^ insertion or when compressions are performed from the manikin’s side (when performing two-responder CPR), than when the over-the-head position is used with an i-gel^®^ in place.

## 2. Methods

In the present study, we performed a post hoc analysis of our original dataset [8], using only data recorded in the i-gel^®^ group. All data were automatically collected through the manikin’s sensors and extracted to a comma-separated values (CSVs) file, thereby preventing assessment bias. In the original study, the variables of interest were automatically generated using a custom-coded PHP script, which had to be adapted for the purpose of the present analysis.

Each entry was divided in two periods (Figure 1). The “Pre-SGA period” was defined as the time between the first compression and the second shock. This time included the compressions performed before the insertion of the SGA device and compressions performed from the manikin’s side. The “SGA period” began after the second shock and lasted until the end of the scenario, which lasted 10 min and during which ROSC could not be achieved. During this period, all the chest compressions were delivered in an over-the-head position while the i-gel^®^ was in place (https://swiss-cpr-studies.ch/cpr2-intro-vid, (accessed on 20 February 2022)).

The primary outcome was the percentage of the compressions with a depth of 5 to 6 cm. The secondary outcomes were the mean compression depth, mean compression rate, percentage of the compressions with a rate of 100 to 120 compressions per minute (cpm), and percentage of correct chest recoil.

The distributions were assessed graphically and by using the Shapiro–Wilk test in case of doubt. A paired *t*-test, a Wilcoxon signed-rank test or a sign test was applied, depending on the assumptions.

## 3. Results

The experimental group consisted of 13 simulations. Among them, 11/13 (84.5%) teams switched to 30:2 CPR, as expected, 2 minutes after the first rhythm analysis; the other teams (2/13, 15.5%) resumed continuous compressions for an additional 2 min before switching to the 30:2 scheme. The mean proportion of the compressions with a correct depth was similar between the pre-SGA and the SGA periods (41.5% (95% CI 23.4–59.6) versus 41.6% (95% CI 25.9–57.4), *p* = 0.99) (Figure 2).

There were no significant differences in the secondary outcomes between the two periods (Table 1).

## 4. Discussion

The shallower compressions obtained from the SGA group did not appear to be linked to their provision in an over-the-head position, while an i-gel^®^ device was in place. While the provision of asynchronous ventilations could probably increase the thoracic cage rigidity in actual out-of-hospital cardiac arrest victims, it is highly doubtful that this was the case in this simulated model, since the cardiac arrest manikin was equipped with a spring designed to allow for a complete recoil. Although the actual cause of these shallower compressions remains unproven, it is probable that they are linked to the introduction of a new airway management device and a new resuscitation approach. Since this hypothesis might also hold true for registry-based studies, we believe that cardiac arrest registries should systematically include the data regarding chest compression quality. Furthermore, we support the use of feedback devices, which have been shown to greatly increase the proportion of adequate chest compressions [9,10]. Consequently, the data related to their use should also be recorded in cardiac arrest registries.

The main limitation of this analysis is the rather limited sample size and the post hoc analysis design. However, an underpowered analysis is unlikely, since a significant difference in the chest compression depths was identified using the same dataset. Moreover, the results are very similar between the two periods and there is a considerable overlap between the confidence intervals.

## 5. Conclusions

In the present post hoc analysis, providing over-the head compressions with an i-gel^®^ device in place had no influence on the chest compression depth or rate, and the reason for the shallower chest compressions found in the i-gel^®^ group remains unknown. The impact of CPR-feedback devices on the chest compression quality when introducing new airway management devices should now be assessed. Finally, it could be valuable to the data regarding chest compression quality more systematically in cardiac registries.

## Figures and Tables

**Figure 1 healthcare-10-00718-f001:**
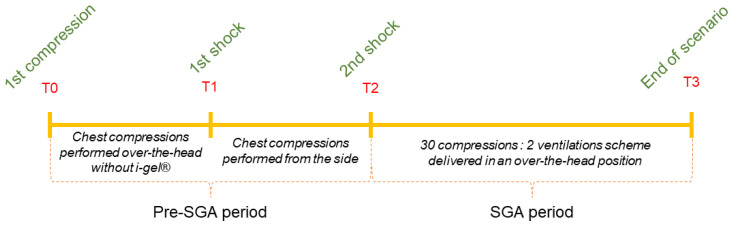
The periods of analysis.

**Figure 2 healthcare-10-00718-f002:**
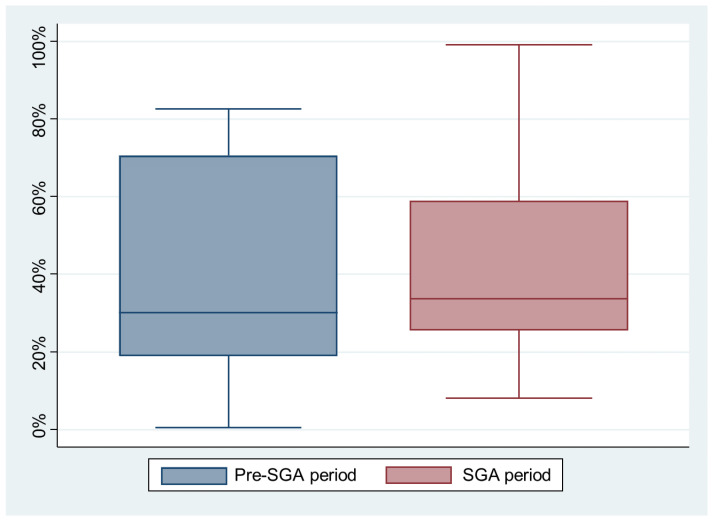
The proportions of the compressions with the correct depths by period (*p* = 0.99).

**Table 1 healthcare-10-00718-t001:** The secondary outcomes.

Outcome	Pre-SGA Period	SGA Period	*p*-Value
Compressions’ depth, mean (95% CI), cm	4.6 (4.3–4.9)	4.5 (4.2–4.8)	0.55
Compressions’ rate, mean (95% CI), cpm	116 (111–121)	116 (112–120)	0.96
Compressions within the rate target, % (95% CI)	73 (55–91)	70 (54–87)	0.47
Compressions with the correct chest recoil, median % (Q1–Q3)	97 (92–99)	100 (86–100)	0.97

## Data Availability

The datasets generated and analyzed during the current study are available in the Mendeley Data repository (https://data.mendeley.com/datasets/s48xv5bbyv/1 (accessed on 20 February 2022)).

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
