# Peer review of "Impact of the Over-the-Head Position with a Supraglottic Airway Device on Chest Compression Depth and Rate: A Post Hoc Analysis of a Randomized Controlled Trial"

_healthcare, 2022, doi:10.3390/healthcare10040718_

Round 1
Reviewer 1 Report
The authors report a post-hoc analysis of a randomized trial comparing the standard protocol of chest compressions with a new approach inserting immediately a supraglottic airway device (i-gel) without chest compression interruption in a simulation CPR. This is an interesting study as bringing new evidence in this field is difficult.
In their original, randomized (13 teams in the standard approach and 13 teams in the experimental approach), simulation study the authors observed significantly higher chest compression fraction in the experimental group but shallower compressions the reason of this post-hoc analysis. According to the results of this analysis shallower compressions in the experimental group are not related to the over-the-had compressions. The mechanism of this difference remain unknown.
Major comments
- In the original work it is explained that in all cases in the experimental group the SGA was inserted immediately. In contrast, in the post-hoc analysis it is not clear the “Pre-SGA period” and “SGA period”. Please if possible in “Methods” provide a flow-chart of events for clarity because is difficult to grasp treatment pathways. Point out the differences and similarities of the arm treatments.
- Please give more information from the original study because is not easy for the reader to link the data. I suggest to summarize the results of the original study in a figure or table to point out other benefits of the new device as it seems that have more drawbacks.
- Please add to the limitations of the study the Post-hoc analysis limitations.
- In line 76 ….The experimental group consisted of 13 simulations. Among them, 11/13 teams
switched to 30:2 CPR, as expected, two minutes after the first rhythm analysis… Please explain the switch of 11 teams.
- Do the authors consider that this new cumbersome device (SGA) could increase the thoracic cage rigidity and make the compressions more difficult. If so it could be added as a probable mechanism.
Minor comments
- To the readers not fresh on the topic please indicate which parameters are more important for high quality compressions in literature and what is the importance of the compression depth.
- Please put the p value in the figure 1.
Reviewer 2 Report
This study presented a post-hoc analysis of a previous study and found no association between the over-the-head position and compression depth.
Data is important, but the study has a sample size limitation. Additional studies with a larger sample must be completed to see if there is no actual difference.
Graphic visualization could be improved following journal guidelines.
